# A Label is Worth a Thousand Images in Dataset Distillation

**Tian Qin**
Harvard University
Cambridge, MA
tqin@g.harvard.edu

**Zhiwei Deng**
Google DeepMind
Mountain View, CA
zhiweideng@google.com

**David Alvarez-Melis**
Harvard University & MSR
Cambridge, MA
dam@seas.harvard.edu

## Abstract

Data *quality* is a crucial factor in the performance of machine learning models, a principle that dataset distillation methods exploit by compressing training datasets into much smaller counterparts that maintain similar downstream performance. Understanding how and why data distillation methods work is vital not only for improving these methods but also for revealing fundamental characteristics of "good" training data. However, a major challenge in achieving this goal is the observation that distillation approaches, which rely on sophisticated but mostly disparate methods to generate synthetic data, have little in common with each other. In this work, we highlight a largely overlooked aspect common to most of these methods: the use of soft (probabilistic) labels. Through a series of ablation experiments, we study the role of soft labels in depth. Our results reveal that the main factor explaining the performance of state-of-the-art distillation methods is not the specific techniques used to generate synthetic data but rather the use of soft labels. Furthermore, we demonstrate that not all soft labels are created equal; they must contain *structured information* to be beneficial. We also provide empirical scaling laws that characterize the effectiveness of soft labels as a function of images-per-class in the distilled dataset and establish an empirical Pareto frontier for data-efficient learning. Combined, our findings challenge conventional wisdom in dataset distillation, underscore the importance of soft labels in learning, and suggest new directions for improving distillation methods. Code for all experiments is available at `https://github.com/sunnytqin/no-distillation`.

## 1 Introduction

Data is central to the success of modern machine learning models, and there is increasing evidence that data *quality* trumps quantity in many settings. For example, in the context of large language models (LLM), Abdin et al. [1], Gunasekar et al. [11], and Hughes [14] show that training LLMs in a "data-optimal" regime allows for significant reductions in model size without sacrificing performance and capabilities. In other words, when trained on high-quality data, small models can rival the performance of their much larger counterparts. Despite its obvious importance, there is no clear answer to what characteristics define "good data." Data distillation offers one approach to answering this question. First introduced by Wang et al. [29], the goal of dataset distillation is to condense a dataset into a smaller (synthetic) counterpart, such that training on this distilled dataset achieves performance comparable to training on the original dataset. By studying the distillation process, we can thus investigate what information is preserved in the data (features, symmetries, and information), which in turn may shed light on fundamental characteristics that make for "good" training data.

The success of dataset distillation raises two key questions. First, what aspects of the training data best facilitate "data-efficient" learning? Second, what distillation procedure better captures the aspects of the dataset relevant for prediction? So far, dataset distillation work has focused almost exclusively

38th Conference on Neural Information Processing Systems (NeurIPS 2024).

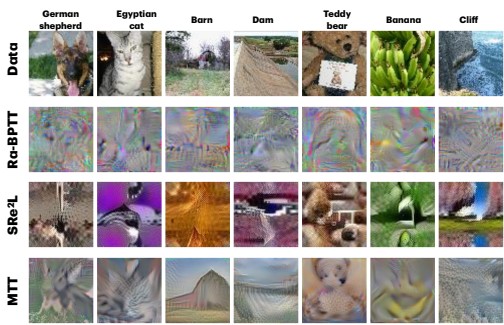
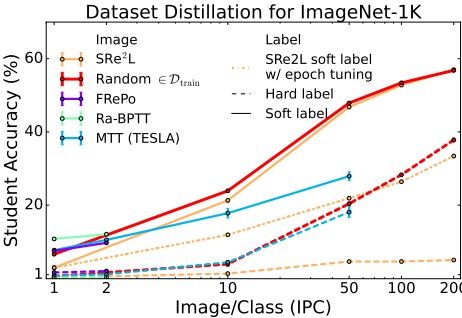

Figure 1: **Soft labels are crucial for dataset distillation** *Left:* Synthetic images by different distillation methods. *Right:* Student test Accuracy comparison between different distillation methods and random baseline from training data (both with hard labels or with soft labels). For soft/hard label generation details, see Appendix A.2.

on the second question, with a strong emphasis on improving ways to generate synthetic data for each class [23]. Figure 1 (left) shows examples of the synthetic images generated by various state-of-the-art (SOTA) data distillation methods. Our first key observation is that, despite very different generation strategies resulting in perceptually distinct synthetic images, virtually all leading methods, especially those able to scale to large datasets such as ImageNet-1K (or its downsized version) ([6, 9, 33, 39]), use soft (i.e., probabilistic) labels for the datasets they distill. Furthermore, Cui et al. [6] and Yin et al. [33] directly use soft labels generated by pretrained experts.

Based on this observation, we take a step back and revisit the first key motivating question above. Specifically, we ask: what is the relative importance of features (e.g., images) and labels for distillation—and therefore, data-efficient training? To answer this question, we design a series of ablation experiments aimed at studying the role of labels in dataset distillation, including a simple but effective baseline that pairs randomly sampled training images (i.e., not synthetic, learned ones) and with soft labels. Surprisingly, we observe that (i) the main factor explaining the success of these methods is the use of soft labels, and (ii) the simple soft label baseline achieves performance comparable to SOTA dataset distillation methods (Figure 1, right).[1] This result alone has important implications for practitioners and researchers alike. For practitioners, it calls into question spending compute or research efforts on generating synthetic images, given their relatively limited contribution to overall distillation performance. For researchers, it suggests revisiting common assumptions about existing data distillation approaches and considering alternatives.

After demonstrating the importance of soft labels, and given the limited analysis of their role in data distillation, we design further experiments to study in detail what makes them so effective for learning. We focus on the setting where soft labels are generated by an "expert" model, e.g., by labeling an image with the class probabilities predicted by the model, as most SOTA distillation methods do. We find that when learning in a data-limited setting, a student model trained on soft-labeled data achieves its best performance by learning from an early-stopped expert, and its success relies on leveraging semantic structure contained in these soft labels. We also present an empirical knowledge-data scaling law along with a Pareto frontier that showcase how (optimal) knowledge can be used to reduce dataset size. Next, we argue that the soft label baseline is a special case of Knowledge Distillation (KD) [13], where the role of soft labels has been better appreciated. Our final set of experiments seek to turn these findings into better methods for generating soft labels. To this end, we first show that expert ensembles can improve learning value of soft labels. We also show that directly optimizing labels with data distillation methods (without training any experts) recovers the *same* information in labels, indicating that expert knowledge might be the *necessary* way for dataset distillation.

Taken together, the results of our in-depth analysis highlight the importance of labels in dataset distillation, challenging conventional wisdom. Our contributions can be summarized as follows:

---

[1]Ra-BPTT [9] and FRePo[39] only scale to IPC=1, 2 on downsized ImageNet-1K. TESLA[6] is an MTT[4] variant with memory-efficient implementation and uses soft labels.

- We conduct a series of ablations, including a simple-but-powerful baseline, to show that the success of existing data distillation methods is driven not by synthetic data generation strategies but by the use of informative labels.
- We show that *structured semantic information* (e.g., distributional similarity across related classes) is key to good soft labels and that different distillation budgets imply different optimal soft label structure profiles. Furthermore, we show how to effectively modulate these profiles by using expert labelers trained with early stopping, a strategy that turns out to recover soft labels obtained with existing data distillation methods as a particular case.
- We establish an empirical data-knowledge scaling law to quantify how knowledge can be used to reduce dataset size, and we also establish a Pareto frontier for data-efficient learning.

## 2    Related Work

**Dataset distillation.** Data distillation methods have been primarily developed for image classification tasks. Existing methods can be organized into three main categories: (1) meta-model matching, (2) distribution matching, and (3) trajectory matching [23]. Meta-model matching methods approach the problem by solving the original bi-level optimization formulation of Wang et al. [29]. To tackle the computation challenge, various methods have been proposed to either approximate the objective [20] or improve the optimization process [9, 15, 39]. Distribution Matching (DM) seeks to minimize the statistical distance in the output (logit) space between original and distilled dataset samples [36]. Further refinements to the method include [16, 28, 35, 37, 39]. Yin et al. [33] propose to match network statistics (batchnorm) along with logits and then assigning soft labels to synthetic images. Follow-up works [12, 32] brings further improvements to the soft label assignment strategy. Matching training trajectories (MTT) was proposed by Cazenavette et al. [5], suggesting that matching long-range training dynamics can serve as a proxy objective to the expensive bi-level formulation. Further improvements and variations include [6, 8]. In this work, we aim to uncover common factors responsible for the success of these methods (i.e., the use of soft labels). We also draw connection between data distillation and knowledge distillation. Concurrent work [30] also discovers that models learned from distilled data behaves similar to an early stopped expert.

**Distillation with soft labels.** Almost all existing distillation methods able to scale to ImageNet-1K (downsized or original) leverage soft labels. Cui et al. [6] and Yin et al. [33] decouple image learning from label assignment and directly use pre-trained experts to assign soft labels. Feng et al. [9] and Zhou et al. [39] learn the distilled images and soft labels simultaneously. In the very early years of data distillation research, the importance of soft labels was highlighted by Bohdal et al. [3] and Sucholutsky and Schonlau [26], although this was done in a very limited experimental setting (MNIST), limiting its influence in further work that tackled larger and more complex classification tasks more representative of modern machine learning. Despite the increasing prevalence of soft labels in state-of-the-art dataset distillation methods, little work has been done in a controlled setting to understand how data and labels each contribute to the quality of the distilled data. This work extensively addresses this question.

**Knowledge Distillation.** The goal of knowledge distillation (KD) is to transfer knowledge learned by a large teacher model to a small student model [2, 10, 13]. While the KD objective may seem at odds with the dataset distillation objective, many dataset distillation methods [32, 33] are directly inspired by data-free KD methods [25, 31]. In fact, the use of soft labels has been extensively studied in the context of KD [13, 19, 27]. Generally, soft labels function both as supervisory signals and regularizers. Yuan et al. [34] and Zhou et al. [38] further argue that under the KD setting the regularization effect from soft labels is equally —if not more— important than sharing knowledge. In this work we uncover deeper connections between the two fields. We show that one way to achieve dataset distillation is to incorporate expert knowledge into soft labels (i.e., KD). We further show that, unlike the conclusions drawn in KD, the knowledge-sharing aspect of soft labels is the dominant effect under the data distillation setting.

## 3    Soft Labels are Crucial for Distillation

### 3.1    Background on dataset distillation and a simple soft label baseline

The goal of dataset distillation is to create a condensed version of a larger dataset that retains the essential information needed for training a model. Formally, we denote the original dataset as

Table 1: **Benchmark SOTA methods against CutMix baseline and soft label baseline on ImageNet-1K.** SRe$^2$L is the only method that can scale to ImageNet-1K. Both soft label-based baselines with random training images can already achieve comparable performances.

| Labeling Strategy | Hard label | Soft label baseline | | | | CutMiX augmented soft labels | |
|---|---|---|---|---|---|---|---|
| Labeling Expert | None | ResNet18 | | ResNet50 | | ResNet18 | |
| Image Generation | Random $\in \mathcal{D}_{train}$ | Random $\in \mathcal{D}_{train}$ | | Random $\in \mathcal{D}_{train}$ | | Random $\in \mathcal{D}_{train}$ | SRe$^2$L |
| Eval Model | ResNet18 | ResNet18 | ResNet50 | ResNet18 | ResNet50 | ResNet18 | ResNet18 |
| IPC=1 | 0.6 (0.1) | 6.6 (0.4) | 6.8 (0.4) | 6.7 (0.4) | 7.3 (0.2) | 1.6 (0.1) | 2.9 (0.5) |
| 10 | 3.9 (0.6) | 23.9 (0.4) | 24.0 (0.4) | 22.9 (0.6) | 23.9 (0.5) | 25.8 (0.7) | 21.3 (0.6) |
| 50 | 20.4 (0.4) | 47.9 (0.6) | 53.1 (0.5) | 43.0 (0.6) | 53.2 (0.3) | 54.3 (0.6) | 46.8 (0.2) |
| 100 | 28.2 (0.3) | 53.4 (0.5) | 57.6 (0.4) | 53.5 (0.5) | 58.3 (0.5) | 54.7 (0.2) | 52.8 (0.3) |
| 200 | 37.8 (0.5) | 56.8 (0.4) | 62.3 (0.6) | 56.5 (0.8) | 62.7 (0.4) | 57.7 (0.6) | 57.0 (0.4) |
| Full | ResNet18 = 69.8% | | | | ResNet50 = 76.1% | | |

$\mathcal{D}_{target} = \{(x_i, y_i)\}$, where $x_i$'s are input images and $y_i$'s are labels. Similarly, we can denote the distilled dataset as $\mathcal{D}_{syn} = \{(\tilde{x}_i, \tilde{y}_i)\}$, and to achieve dataset size reduction $|\mathcal{D}_{syn}| \ll |\mathcal{D}_{target}|$. Dataset distillation problem can be formulated as a bi-level optimization problem [29]:

$$\underset{\mathcal{D}_{syn}}{\arg\min} \mathcal{L}(\theta^{\mathcal{D}_{syn}}; \mathcal{D}_{target}) \quad \text{s.t.} \quad \theta^{\mathcal{D}_{syn}} = \underset{\theta}{\arg\min} \mathcal{L}(\theta; \mathcal{D}_{syn}) \tag{1}$$

By convention, the size of the distilled data set is often quantified in terms of images per class (IPC). The goal of this work is to study the role of $\tilde{y}$ in a controlled setting while fixing $\tilde{x}$.

Our first set of experiments comparing the performance of popular distillation methods with hard vs. soft labels (Figure 1) show that soft labels are crucial for the performance of those distillation methods. On the other hand, the use of pretrained experts during distillation is also common. These two observations suggest an ablation study to evaluate the importance of soft labels in the context of dataset distillation. To this end, we introduce a simple baseline that "distills" datasets by selecting *real* (i.e., not synthetic) samples and soft-labeling them. Specifically, we randomly sample images from the training data and use pretrained experts to generate labels for them. The only hyper-parameter is thus which expert to use. To generate diverse experts, we save checkpoints at each epoch and vary which checkpoint is used based on the data budget.

### 3.2 Benchmarking distillation methods against the soft label baseline

From the many dataset distillation methods that have been proposed, we select the top-performing one from each of the three categories summarized in Section 2 as a representative: Ra-BPTT [9] (bi-level optimization objective), MTT [4] (trajectory matching objective), and SR2$^2$L [33] (distribution matching objective). Among these, SRe$^2$L[33] is the only one that generalizes to ImageNet-1K without the need to downsize it (see Appendix A.3 for downsized comparison). SRe$^2$L first generates images with a distribution matching objective using a pretrained expert. The same expert is then used to generate soft labels for these synthetic images. Specifically, SRe$^2$L proposes generating 300 soft labels per prototype, where each soft label corresponds to a unique augmentation to the image. We establish an additional baseline, which we denote as "CutMix augmented soft labels" since SRe$^2$L directly derives the labelling method from from Shen and Xing [25]. In this additional baseline, we remove the synthetic image generation process and replace it with random images sampled from the training data. We then employ the same CutMix augmented label generation. Table 1 shows that randomly sampled training images paired with soft labels yield performance comparable to distilled synthetic ones. Moreover, the storage cost of the 300 soft labels needed for SRe$^2$L is at least 300$\times$ larger than that of the soft label baseline [22]. We also observe that when the labels are generated by a teacher model with a different architecture, the student still performs well. In other words, the soft label baseline meets the cross-architecture generalizability requirement for dataset distillation.

In addition to ImageNet, we benchmark the performance of these methods on smaller datasets, where they generally tend to perform better. We compare the soft label baseline against other methods on TinyImageNet, CIFAR-100, and CIFAR-10, as shown in Table 2. For these datasets, existing methods sometimes significantly outperform the baseline at very low data budgets (i.e., high compression rates). However, at higher data budgets, the soft label baseline again yields better or comparable performances. These results show that current methods still face issues scaling to larger data budgets.

Table 2: **Benchmark SOTA methods against soft label baseline ("Sl baseline") on TinyImageNet, CIFAR-100 and CIFAR-10.** For smaller datasets at small data-budget, SOTA distillation methods can outperform the baseline. Scaling those methods to large data-budget remains a challenge.

| Dataset | TinyImageNet | | | | CIFAR-100 | | | | CIFAR-10 | | | |
|---|---|---|---|---|---|---|---|---|---|---|---|---|
| **Distill Method** | Ra-BPTT | MTT | DM | Sl Baseline | Ra-BPTT | MTT | DM | Sl Baseline | Ra-BPTT | MTT | DM | Sl Baseline |
| IPC=1 | 20.1 (0.3) | 8.8 (0.3) | 3.9 (0.2) | 7.6 (0.3) | 35.3 (0.4) | 24.3 (0.3) | 11.4 (0.3) | 16.0 (0.3) | 53.2 (0.7) | 46.3 (0.8) | 26.0 (0.8) | 21.1 (0.5) |
| 10 | 24.4 (0.2) | 23.2 (0.2) | 12.9 (0.4) | 27.2 (0.4) | 47.5 (0.2) | 40.1 (0.4) | 29.7 (0.3) | 34.3 (0.2) | 69.4 (0.4) | 65.3 (0.7) | 48.9 (0.6) | 46.3 (0.8) |
| 50 | - | 28.0 (0.3) | 24.1 (0.3) | 35.6 (0.4) | 50.6 (0.2) | 47.7 (0.3) | 43.6 (0.4) | 47.1 (0.4) | 75.3 (0.3) | 71.6 (0.2) | 63.0 (0.4) | 62.1 (0.5) |
| 100 | - | - | - | 38.2 (0.2) | - | - | - | 51.3 (0.5) | | | | |
| Full | ConvNet(F) = 38.5% | | | | ConvNet(F) = 56.4% | | | | ConvNet(F) = 81.5% | | | |

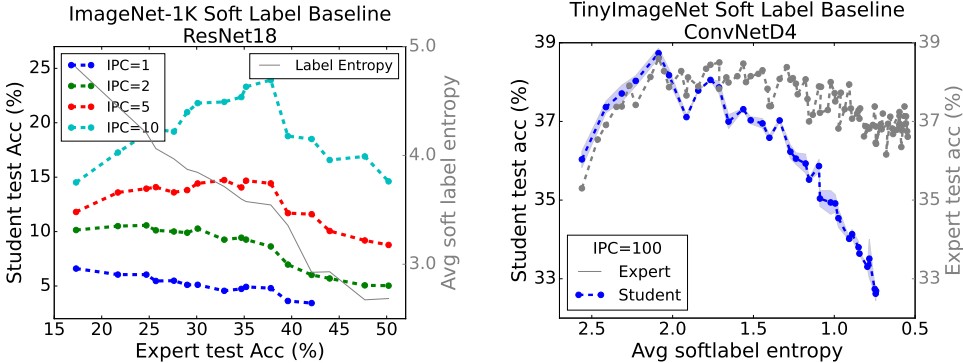

Figure 2: **Expert test accuracy v.s. student test accuracy v.s. soft label entropy.** The quality of soft labels (measured by student accuracy) depends on expert accuracy (*left*) and label entropy (*right*).

Importantly, the soft label baseline is robust to random image selection and expert training seeds (Appendix A.4). Using a simple image selection strategy (cross-entropy) could improve the performance of the soft label baseline (Appendix B). However, the goal of establishing this baseline is not to achieve the best performance but to showcase the importance of soft labels for dataset distillation. In the next section, we aim to understand why soft labels are so important for learning from limited data.

# 4 Good Soft Labels Require Structured Information

Despite its embarrassing simplicity, the soft label baseline yields strong performances, suggesting that labels are an important —perhaps the most important— aspect of data-efficient learning. Therefore, we next seek to understand why soft labels are so effective for learning. In Section 4.1, we start with a generation observation that expert epoch determines the optimal soft label. In Section 4.2, we develop an understanding that the structured information encoded in soft labels is what allows models to learn from so little data. Finally, in Section 4.3, we establish an empirical knowledge-data scaling law and an extreme version of distillation — learning with no (feature) data.

## 4.1 General observations from the soft label baseline

In establishing the baseline, we observe that the optimal expert to generate soft labels depends on the (distilled) data budget. Specifically, for smaller data budgets, it is often better to use labels generated by an early-stopped expert, which we will refer to "expert at epoch $x$." As an example, Figure 2 visualizes how using expert at different epochs impact the student test accuracy for ImageNet-1K.

As we train the expert for more epochs, two things in soft labels can change: the information being conveyed in softmax values (i.e., the softmax distribution itself), and the amount of information being conveyed (i.e., the entropy). In Figure 5, we visualize the expert soft labels for a randomly selected image from TinyImageNet (a German Shepherd, with the corresponding label shown in red) at 4 different epochs. At epoch 0, the soft labels are randomly initialized. The expert at epoch11 (optimal for IPC=1) considers "Bison" as top guess but only assigns $2\%$ probability. At epoch 20, the top two guesses become "Brown bear" and "Bison." Finally, at epoch 50 (roughly optimal for IPC=10), the expert correctly assigns "German Shepherd" with the maximum likelihood. This anecdotal example shows that the information being conveyed can change over the course of expert

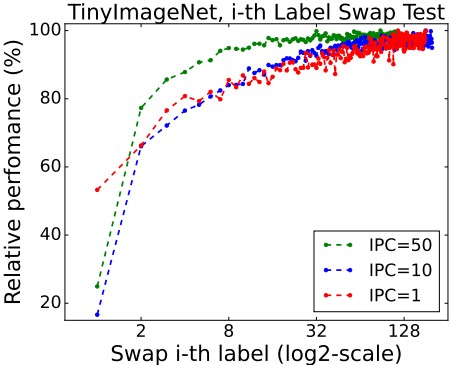

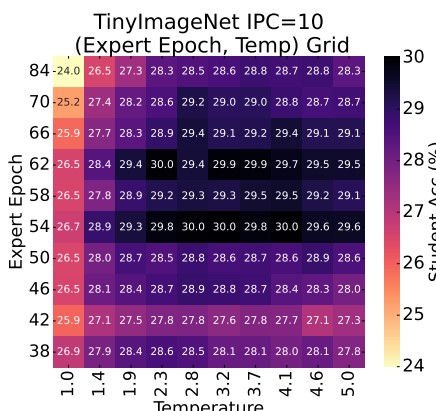

Figure 3: **Importance of $i$-th label by performing label swapping test.** Swap the $i$-th label (sorted by softmax value) with the last label. Top labels contain structured information and the non-top labels contain noise.

Figure 4: **(Expert Epoch, Temp) grid search on TinyImageNet IPC=10.** Temperature smoothing does not fully resolve the issue that later epoch experts yield sub-optimal labels for a given data budget.

training. Besides changes in the per-class information, the entropy of labels also drop over the course of training. Eventually, the labels sharpen and converge to hard labels, meaning less information is being conveyed. As a first attempt to isolate these two factors (information, entropy), we look at experts that have been trained until convergence. Figure 2 (right) shows that the expert starts to converge and even slightly overfits. The test accuracy stabilizes around $38\%$, but the entropy of the labels continues to drop drastically. Despite similar performance by the expert, the later epoch experts generate labels that are significantly worse for the student model than the earlier epoch experts. In the next section, we further disentangle the contributions of these two sources of variation (information and entropy) to the quality of soft labels.

## 4.2 Structured information in soft labels and its importance for distillation

We first distinguish two components in soft labels - *structured information* and *unstructured noise*. We define structured information as softmax values that the student models learn from - they contain information such as semantic similarities between classes (see Appendix C.1 for a visualization of information in soft labels). Unstructured noise, on the other hand, contains no information but may provide regularization during student training. The contribution of these two effects has only been studied in KD settings [34, 38]. In data distillation, we aim to quantify the percentage of softmax values that provide learnable information and the percentage that contains only noise. We also want to determine whether the optimal structured information is unique to each data budget.

**Label swapping test.** To quantify the percentage of softmax values that contain structured information, we design a "$i$-th label swapping test." This experiment relies on the assumption

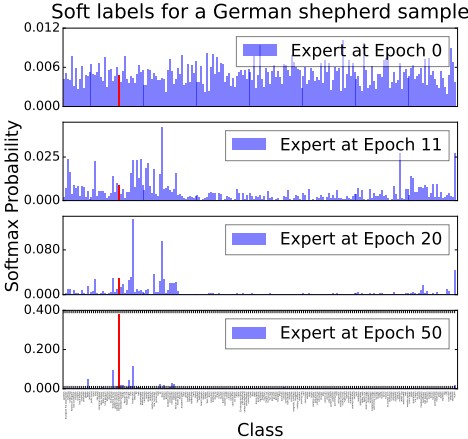

Figure 5: **Soft labels generated by expert at different epochs.** The structured information in soft labels changes over the course of training.

that if we sort labels by their softmax values in descending order, the last label likely contains no useful information. In the experiment, we swap $i$-th label with the last label, and measure the relative performance compared to unswapped labels. Refer to Appendix C.2 for a detailed experiment procedure. Figure 3 shows $i$-th label swapping test results on TinyImageNet. For all IPC budgets, swapping top labels significantly hurts the performance, indicating when learning with few data,

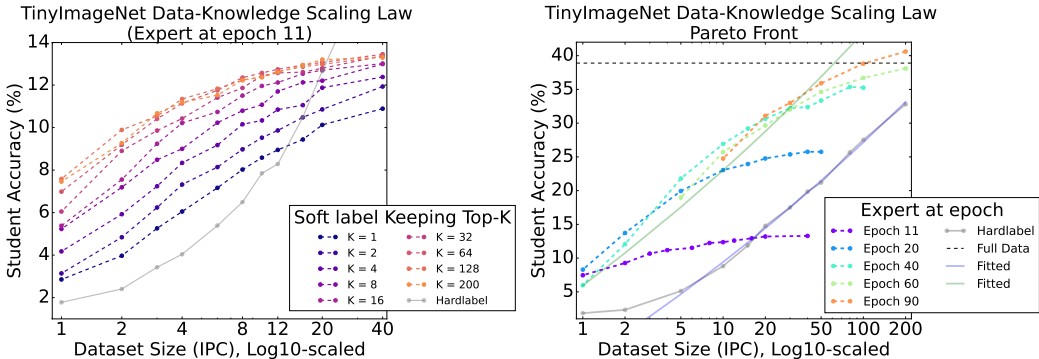

Figure 6: **Data-Knowledge Scaling Law for TinyImageNet.** *Left*: Trading knowledge (amount of information in soft labels) with data using one expert. *Right*: Establishing the Pareto-optimal front for data-efficient learning.

students rely more on structured information (knowledge) during learning. For 1 IPC, the top label does not contain as much information compared to higher IPC. It is likely because the top-1 label generated by an early-epoch expert, which itself has poor performance, does not contain much useful information.

**Effect of temperature and epoch in soft labels.**    So far we have only observed that for a given data budget, there is an optimal expert epoch for label generation (Figure 6). This observation does not imply that the structured information generated by this expert is indeed optimal. Specifically, we could not eliminate the possibility that labels generated by the expert at a later epoch may contain better information but too sharp for the student to learn from (i.e., suboptimal entropy). To further disentangle the two factors (information and entropy) and understand the impact of label entropy on label quality, we use a solution from KD: varying the temperature of the softmax value [13] to control label entropy. Specifically, we perform an extensive grid search on (Expert epoch, Temperature) under TinyImageNet IPC=10 setting, results shown in Figure 4. Increasing the temperature *does* further improve the soft label baseline performance, but the optimal epoch in the temperature=1 setting remains optimal even when temperature smoothing is used. From these two experiments, we conclude that students leverage structured information in soft labels during learning, and the optimal information (expert) is unique to each data budget.

### 4.3   Trading data with knowledge

We have established that the smaller the data budget, the more heavily the student model relies on structured information present in the soft labels. In this section, we further explore the trade-off between data and knowledge in two settings. First, we explore a scaling law to quantify how knowledge can reduce dataset size. Second, we further the idea of "learning from limited data" to "learning from no data" for a given class.

**Scaling law.**    The data-knowledge scaling law experiment quantitatively measures how knowledge can reduce dataset size *and* the optimal knowledge for each dataset size. For the first experiment, we use an expert trained on TinyImageNet for 7 epochs (with a test accuracy of $11\%$) as the teacher model. As in the baseline setting, the data (images) are randomly selected from the original training set. To establish the empirical scaling law, we vary dataset sizes (measured by IPC) and the amount of knowledge in the soft labels. To control the amount of knowledge in the soft labels, we retain different top-$k$ values (sorted by softmax probabilities in descending order) and zero out the rest *without* re-normalization. We compare the data-knowledge scaling law against the standard training scaling law (i.e., using hard labels from data).

The results of the knowledge-data trade-off experiment are shown in Figure 6 left. To fully recover the teacher model, the student needs around 10 IPC if full knowledge is present ($k$=200). But the IPC value quadruples to 40 if we reduce $k$ to 32. Moreover, learning from this expert becomes suboptimal as the dataset size increases, and students trained on hard labels start to outperform. This finding supports the observation from Section 4.2: with an increased data budget, the student benefits more from learning from a model at a later epoch. In Appendix C.3, we repeat the knowledge-data trade-off

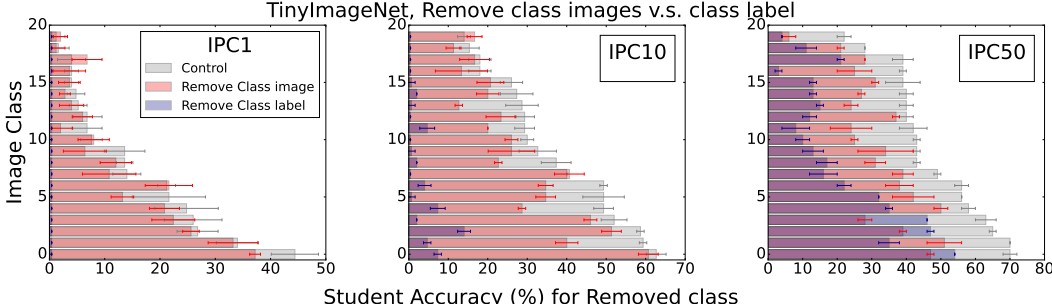

Figure 7: **Zero-shot learning in the absence of knowledge v.s. data.** The student model can achieve good performances when data is absent but much worse performances when knowledge is absent.

experiment using an expert trained for later epochs. By combining experts at different epochs with different data budgets, we can establish a Pareto-optimal front for data-efficient learning, as shown in Figure 6 right, and curve fit below. The use of expert knowledge provides a constant $6\times$ data size reduction since $|\mathcal{D}| = \text{num class} \times \text{IPC}$:

$$\text{Hard Label: Student Test Acc} = \left(\frac{\text{IPC}}{29.9}\right)^{0.077} - 0.8$$

$$\text{Soft Label: Student Test Acc} = \left(\frac{6.04 \times \text{IPC}}{29.9}\right)^{0.077} - 0.8$$

**Learning from (almost) no data.** We have used the empirical knowledge-data scaling law experiment to establish that one can directly trade knowledge with data. This observation inspires us to push this idea to the extreme. For a selected class, in which of the following two scenarios would the student perform better? 1) The student learns in the complete absence of data from this class. 2) The student learns with data from this class but in the absence of knowledge.

For a selected class $i$, we design a zero-shot experiment with a control and two treatment groups:

- Control: The student model is trained with data from all classes and full soft label knowledge.
- Treatment 1 - Remove image: We remove *all* training images from class $i$ while keeping the softmax values corresponding to class $i$ in the rest of the training data.
- Treatment 2 - Remove label: We keep images from class $i$ and the *full* soft label for those images. For all other images in the training data that don't belong to class $i$, we zero-out softmax values corresponding to class $i$ *without* re-normalization.

Figure 7 shows the experimental results on TinyImageNet under three IPC settings (classes sorted by control group performance). Compared to the control group, the performance of the remove-data treatment suffers only a little.This good performance implies that the student can perform zero-shot classification on the removed class *without* having seen any images from the class. In contrast, the student's performance suffers drastically when labels are removed. For IPC=1, the student model fails to learn anything at the remove-label setting ($0\%$ test accuracy for removed classes), and the performance of the remove-label treatment only starts to catch up at IPC=50.This substantial performance difference between the two treatment groups suggests that the information conveyed in labels is more important than the data.

## 5 Soft Labels are Not Created Equal

So far we have established that one way to achieve dataset distillation is by trading knowledge for data, specifically using knowledge from pretrained experts. In this section, we continue to work under the setting where images are randomly sampled from the training data, but we expand on different ways to obtain "knowledge." In Section 5.1, we show that an expert ensemble can consistently improve soft label quality. In Section 5.2, we discuss the possibility of obtaining soft labels through data distillation methods.

Table 3: **Comparison of label generation strategies on TinyImageNet.** Ensemble provides consistent performance improvement. Distillation method (BPTT) can be adopted to learn labels.

| Label Source | Data | Expert | | Distilled | |
|---|---|---|---|---|---|
| Labeling Method | Hard | Single | Ensemble | BPTT | MTT |
| IPC=1 | 1.6 (0.1) | 7.6 (0.3) | 8.7 (0.1) | 13.5 (0.3) | 2.2 (0.0) |
| 10 | 7.3 (0.2) | 27.2 (0.4) | 30.1 (0.2) | 25.0 (0.4) | 10.3 (0.2) |
| 50 | 20.7 (0.3) | 35.6 (0.4) | 39.3 (0.4) | - | 22.2 (0.5) |

## 5.1 Expert ensemble

A common strategy to boost predictive performance in machine learning is to use ensemble predictors. By combining predictions from various models, ensemble methods effectively capture a more comprehensive representation of the underlying data distribution. Therefore, we hypothesize that averaging soft label predictions from multiple experts might improve the quality of the labels. We train 10 ConvNet experts with different random seeds on TinyImageNet. To generate ensemble soft labels, we average the logits from all the experts for each of the randomly selected training images before passing them through the softmax calculation. For a fair comparison, we use the expert epoch previously determined in the soft label baseline. Table 3 shows that soft labels derived from expert ensembles consistently improve student model accuracy compared to single-expert labels.

## 5.2 Learning soft labels through data distillation methods

Instead of using pretrained experts to generate soft labels, a natural alternative is to learn labels through dataset distillation methods. The motivation is twofold: First, we aim to explore different approaches in hopes of finding better ways to obtain labels. More importantly, we have demonstrated that using the knowledge-distillation framework (i.e., using pretrained experts) is a *sufficient* mechanism to obtain informative labels. We now want to understand whether the information generated by experts is the only (or *necessary*) solution.

Distribution matching-based distillation methods, one of the three predominant categories of successful distillation methods identified above, cannot be easily adapted to learn labels because they aim to generate synthetic images that minimize cross entropy on a teacher model. This leaves two other possible families of methods: training trajectory matching and bi-level optimization. For both methods, we can adopt the same training objective but freeze the images and learn the labels only. For the training trajectory matching objective, we adopt MTT [4], and for the bi-level optimization objective, we adopt truncated-BPTT [9] (also known as BPTT). We experiment with these two adaptations on TinyImageNet, with results shown in Table 3. The MTT adaptation fails to meaningfully improve on the hard label baseline—see Appendix D.1 for a detailed description of the MTT objective and a discussion on why it fails. On the other hand, the BPTT adaptation yields meaningful results, even slightly improving on ensemble labels in the IPC=1 setting.

To understand whether expert knowledge is *necessary* to obtain useful soft labels, we compare BPTT-learned labels with ensemble-generated labels. The motivation behind comparing BPTT labels is not due to their performance, but because BPTT addresses dataset distillation directly from its problem formulation (Eqn. 1). By directly solving the bi-level optimization problem, no experts are trained during the distillation process. Moreover, it eliminates concerns about whether a proxy distillation objective, such as training trajectory matching, may introduce bias into the distillation outcome. See Appendix D.2 for a detailed description of truncated-BPTT algorithm and our adaptation to learn labels.

We fix the randomly sampled training images and generate soft labels using an expert ensem-

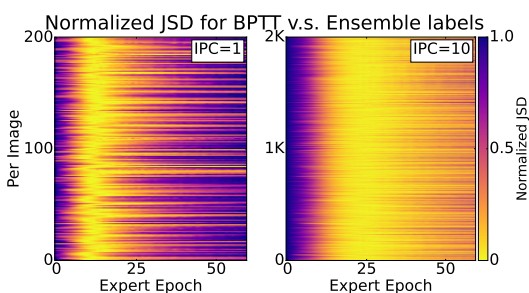

Figure 8: **Normalized JSD between BPTT-learned labels and ensemble-expert labels on TinyImageNet.** Despite not training any experts, distillation method (BPTT) recovers the same labels as those generated by early stopped experts.

ble and using BPTT. We use ensemble labels instead of a single expert labels to reduce bias. We use the Jensen-Shannon Distance (JSD) to quantify the distance between the softmax distributions. For each image, we also perform a min-max normalization across all expert epochs to identify the epoch at which the BPTT-generated label for that image is most similar to the ensemble-generated labels. See Appendix D.2 for a detailed description of the comparison methodology. Figure 8 shows the normalized JSD for all images under the TinyImageNet IPC=1 and IPC=10 settings. For both data budgets, the minimizing epoch for the normalized JSD across all images is roughly the same (Epoch 13 for IPC=1, 25 for IPC=10). Conveniently, epoch 13 is fairly close to the optimal early-stop epoch for IPC=1 (optimal stop epoch = 11). In Appendix D.2, we also compare raw JSD values to further ensure that the distributions are sufficiently close to each other on an absolute scale.

The two observations above provide strong evidence that BPTT recovers the same labels generated by the optimal early-stopped expert, *without* explicitly training any. Note that by restricting to only learning labels, we limit the scope of the dataset distillation problem. However, in this restricted setting, the dataset distillation solution converges to the knowledge distillation solution (i.e., labels from an optimal early-stopped expert), indicating that expert knowledge is *necessary* to obtain informative labels. A future direction is to explore how this conclusion might change when images are learned simultaneously with labels.

# 6   Discussion and Conclusions

This paper highlights the crucial role of soft labels in dataset distillation. Through a series of ablation experiments and a simple soft label baseline, we have shown that the use of soft labels is the main factor behind the success of state-of-the-art dataset distillation methods. We also demonstrated that structured information in soft labels is crucial for data-efficient learning. We established an empirical knowledge-data scaling law to characterize the effectiveness of using knowledge to reduce dataset size and an empirical Pareto frontier for data-efficient learning. Additionally, we showed that when only learning labels, data distillation converges to the knowledge distillation solution.

**Implications for dataset distillation**   Our findings have several implications. First, they suggest rethinking existing data distillation methods that emphasize on synthetic image generation techniques, while under emphasize the importance of informative labels. Shifting the focus of distillation research to account for both of these factors is likely to lead to novel, potentially better, approaches. For example, in this work we only considered the setting where images are randomly selected from the training data, but future work may explore simultaneous image/label synthesis methods to improve distillation performance. Future research may also investigate whether data distillation can be achieved without using expert knowledge, either implicitly or explicitly, in a model-agnostic or even task-agnostic way. For example, future research may investigate how to synthesize images or labels by summarizing characteristics of the dataset, such as symmetries and other invariance priors.

**Limitations and future work**   We have emphasized the importance of soft labels by implementing a simple soft label baseline. While this approach performs well on ImageNet-1K, advanced methods such as [12, 24] continue to push the state of the art. This suggests that, while labels are crucial, successful dataset distillation requires leveraging both images and labels to achieve optimal compression. Future work should explore how best to optimize both components during distillation and examine how each uniquely influences student model learning. Additionally, investigating what other types of information, beyond expert knowledge, can be distilled for effective compression remains an open research question.

On the label generation front, we have explored strategies based on existing methodologies, including pretrained experts and Ra-BPTT. Future research could further investigate more effective ways to generate informative labels that enhance model performance.

Lastly, like much of the dataset distillation research, our focus has primarily been on image classification tasks. Although dataset distillation has demonstrated strong results in this area, extending these methods to other tasks and data modalities remains under-explored. While we believe our conclusions are generalizable, a limitation of this work is the narrow range of tasks evaluated. Future studies should explore how distillation techniques perform across a broader set of domains and modalities.

## Acknowledgments

Tian Qin and David Alvarez-Melis were partially supported by the Kempner Institute, the Aramont Fellowship Fund, and the FAS Dean's Competitive Fund for Promising Scholarship. We would also like to thank Clara Mohri and Eran Malach for their insightful discussions and helpful feedback throughout the development of this work. Their input has been invaluable in shaping the direction of our research.

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

# A  Soft label baseline

## A.1  Experimental Details

**Models**   For datasets including downsized ImageNet-1K, TinyImageNet, CIFAR-100, and CIFAR-10 we employ the standard ConvNet architecture that is used by data distillation methods we benchmark against [4, 7, 9, 36]. We use 3 convolutional blocks for low resolution datasets (res:$32 \times 32$, CIFAR-10, CIFAR-100) and 4 convolutional blocks for medium resolution datasets (res: $64 \times 64$, TinyImageNet, downsized ImageNet-1K). For the original size ImageNet-1K, we use the PyTorch [21] official implementation of ResNet18 and ResNet50 to standardize comparison against the SOTA methods [33] we benchmark against.

**Expert training**   We follow a standard training recipe to train experts on downsized ImageNet-1K, TinyImageNet, CIFAR-10, and CIFAR-100. This standard training recipe involves an SGD optimizer and a simple step learning rate schedule, and is used by distillation methods that require experts training in the distillation pipeline [4, 6]. Note that due to the simplicity of the training recipe and model architecture, the expert performance on the full dataset does not reach SOTA image classification performances. The expert training procedure is chosen to facilitate fair comparisons in the context of dataset distillation. For ImageNet-1K, SOTA methods [33] we benchmark against leverage PyTorch pretrained ResNet18 as their teacher model, therefore we adopt the PyTorch official training recipe [17] for expert training.

**Evaluation**   To train student models on distilled dataset, we only allow learning rate tuning and we train student models until convergence. We train five student models with different random seeds and report mean and variance. For other methods we reported in Table 1, 4, and 2, we report values from the original paper. For methods that we report performance without using soft labels (Figure 1 right), we use results reported in the original paper for MTT (TESLA)[6], and FRePo [39]. For Ra-BPTT [9] and SRe$^2$L [33], since they do not report results without soft labels, we acquire the distilled datasets and replace soft labels with hard labels. We perform the same evaluating procedure as above.

**Compute**   All experiments are conducted on NVIDIA A100 SXM4 40GB or NVIDIA H100 80GB HBM3. To train small datasets, experiments typically require less than 5 GPU hours and for large datasets, experiments require at most 100 GPU hours.

## A.2  Label generation and epoch tuning for existing methods

We provide further details regarding how labels are generated in Figure 1(right). We compare our soft label baselines to four existing methods, each with their own soft label generation strategies proposed by the original authors. We apply the original soft label generation strategy used by each method. Overall, some of these methods already include epoch tuning (MTT/TESLA), while others do not (SRe$^2$L), and for some, the concept of epoch tuning is not well-defined (Ra-BPTT and FRePo). In addition, we also compare the performance of the four proposed methods with or without soft labels. We clarify how hard labels and soft labels are obtained for each of the methods:

**SRe$^2$L**   The original method uses labels generated from a fully trained expert without epoch tuning. In our reported results for SRe$^2$L, we include a version of the soft label generation method that directly replicates the original method (detailed in Section 3.2). We also include a version of the soft label generation method that is identical to the soft label baseline, which includes epoch tuning (detailed in Section 3.1). To obtain the hard label counterpart, since SRe$^2$L initializes its distilled images directly from training images, we use the original label of those images as the hard label.

**MTT (TESLA)**   In the original method, the expert used to generate labels is already epoch-tuned, and the epoch also impacts the image generation pipeline. Therefore, we replicate the soft label generation strategy in the original method. To obtain the hard label counterpart, since MTT/TESLA initialize their distilled images directly from training images, we use the original label of those images as the hard label.

**Ra-BPTT**   The original method generates labels along with images using a bi-level optimization objective, so no experts are trained during the process. Because labels are not generated by experts, epoch tuning is not applicable. We use the soft labels generated by the original method. In addition, we have experimented with using pre-trained experts to generate labels for Ra-BPTT generated images but observed that experts trained on real images perform poorly on images synthesized by Ra-BPTT. This is likely because the generated images are too out-of-distribution for experts trained on real training data. Thus, the original labels generated by Ra-BPTT should be considered optimal

for this method. Since Ra-BPTT images are initialized with random Gaussian noise, we use the argmax as hard labels.

**FRePo**   Similar to Ra-BPTT, FRePo is based on Back-Propagation in Time (BPTT), and no experts are trained during the distillation process. Like Ra-BPTT, labels are learned during the distillation process along with images. We use the hard label results reported in the original paper.

### A.3   Downsized ImageNet-1K results

There are three other methods that can scale to ImageNet-1K but a downsized version (reducing image resolution from $256 \times 256$ to $64 \times 64$) and only at small IPC values. We compare their performances against the soft label baseline in Table 4. Among those methods, only Ra-BPTT can slightly outperform the soft label baseline, albeit by a small margin. Scaling these methods to larger data-budget remains a challenge, and as a result, we truncate Table 4 to IPC=2.

Table 4: **Benchmark SOTA methods against the soft label baseline ("Sl Baseline") on (downsized) ImageNet-1K.** Other distillation methods can only scale to downsized ($64 \times 64$) ImageNet-1K and they do not significantly outperform the baseline.

| | ImageNet-1K (Downsized) | | | |
|---|---|---|---|---|
| **Distillation Method** | Ra-BPTT | FRePo | DM | Sl Baseline |
| IPC=1 | 10.77 (0.4) | 7.5 (0.5) | 1.5 (0.1) | 8.0 (0.4) |
| 2 | 12.05 (0.5) | 9.7 (0.2) | 1.7 (0.1) | 9.6 (0.3) |

### A.4   Variance from expert training seed and random image seed

The soft label baseline involves randomly selecting images from training data and use pretrained single expert to generate soft labels for the selected images. We use 6 different random seeds for image selection to confirm that the soft label baseline is not sensitive to random image selection, shown in Table 5. Additionally, we confirm that the soft label baseline is no sensitive to experts training seeds. Specifically, we train six different experts using the same hyperparameters and only varying random seeds. In expert seed experiment, we fix the randomly selected images for the distilled dataset. Results shown in Table 6. As above, student test accuracy is reported as the average of five runs with different random seeds.

Table 5: **Image selection with different random seeds on TinyImageNet.** Our soft label baseline is not sensitive to random image selections.

| IPC | Seed 1 | Seed 2 | Seed 3 | Seed 4 | Seed 5 | Seed 6 |
|---|---|---|---|---|---|---|
| 1 | 7.7 (0.2) | 7.5 (0.2) | 7.3 (0.3) | 7.0 (0.2) | 7.2 (0.2) | 7.5 (0.1) |
| 10 | 26.8 (0.1) | 26.8 (0.2) | 27.3 (0.1) | 26.6 (0.1) | 26.9 (0.2) | 27.0 (0.1) |
| 50 | 36.1 (0.3) | 35.6 (0.5) | 35.6 (0.4) | 36.0 (0.1) | 35.8 (0.3) | 35.6 (0.1) |

Table 6: **Image selection with different expert training seeds on TinyImageNet.** Our soft label baseline is not sensitive to experts with different training seeds

| IPC | Seed 1 | Seed 2 | Seed 3 | Seed 4 | Seed 5 | Seed 6 |
|---|---|---|---|---|---|---|
| 1 | 7.6 (0.1) | 7.5 (0.1) | 8.0 (0.2) | 7.5 (0.2) | 7.9 (0.1) | 7.6 (0.1) |
| 10 | 26.4 (0.4) | 26.8 (0.3) | 26.4 (0.3) | 26.6 (0.2) | 26.4 (0.1) | 26.8 (0.2) |
| 50 | 35.7 (0.1) | 35.9 (0.2) | 36.0 (0.1) | 35.7 (0.4) | 35.7 (0.3) | 36.2 (0.4) |

### A.5   Hyperparameters for expert training and soft label generation

We include the optimal expert epochs to reproduce results reported in Table 1 and Table 2. For smaller IPC values, we train experts with a reduced learning rate so that when we save expert checkpoints, we get expert accuracy at a more granular scale. Note that using a smaller learning rate for expert training is for implementation simplicity. Alternatively, one could use partial epoch checkpoints to achieve the same outcome.

Table 7: **Hyperparameter list to reproduce soft label baseline results in Table 1 and Table 2.**

| Dataset | Expert Architecture | IPC | Expert LR | Student LR | Expert Epoch | Expert Test Accuracy (%) |
|---|---|---|---|---|---|---|
| ImageNet | ResNet18 | 1 | $5 \times 10^{-4}$ | $1 \times 10^{-3}$ | 11 | 17.3 |
| | | 10 | $5 \times 10^{-4}$ | $5 \times 10^{-3}$ | 43 | 35.8 |
| | | 50 | $1 \times 10^{-1}$ | $5 \times 10^{-3}$ | 30 | 60.9 |
| | | 100 | $1 \times 10^{-1}$ | $5 \times 10^{-3}$ | 34 | 62.8 |
| | | 200 | $1 \times 10^{-1}$ | $2 \times 10^{-3}$ | 34 | 62.8 |
| ImageNet | ResNet50 | 1 | $5 \times 10^{-4}$ | $1 \times 10^{-3}$ | 9 | 15.4 |
| | | 10 | $5 \times 10^{-4}$ | $5 \times 10^{-3}$ | 24 | 31.0 |
| | | 50 | $1 \times 10^{-1}$ | $5 \times 10^{-3}$ | 30 | 66.6 |
| | | 100 | $1 \times 10^{-1}$ | $5 \times 10^{-3}$ | 32 | 68.6 |
| | | 200 | $1 \times 10^{-1}$ | $2 \times 10^{-3}$ | 61 | 73.3 |
| TinyImageNet | ConvNetD4 | 1 | $1 \times 10^{-2}$ | $1 \times 10^{-2}$ | 11 | 19.2 |
| | | 10 | $1 \times 10^{-2}$ | $1 \times 10^{-1}$ | 50 | 35.0 |
| | | 50 | $1 \times 10^{-2}$ | $1 \times 10^{-1}$ | 90 | 38.1 |
| | | 100 | $1 \times 10^{-2}$ | $1 \times 10^{-1}$ | 102 | 38.8 |
| CIFAR-100 | ConvNetD3 | 1 | $1 \times 10^{-2}$ | $1 \times 10^{-2}$ | 13 | 33.1 |
| | | 10 | $1 \times 10^{-2}$ | $1 \times 10^{-2}$ | 36 | 47.6 |
| | | 50 | $1 \times 10^{-2}$ | $1 \times 10^{-2}$ | 125 | 52.7 |
| | | 100 | $1 \times 10^{-2}$ | $1 \times 10^{-2}$ | 125 | 52.7 |
| CIFAR-10 | ConvNetD3 | 1 | $5 \times 10^{-4}$ | $1 \times 10^{-2}$ | 20 | 56.0 |
| | | 10 | $5 \times 10^{-4}$ | $1 \times 10^{-2}$ | 44 | 65.1 |
| | | 50 | $5 \times 10^{-4}$ | $1 \times 10^{-2}$ | 128 | 74.9 |
| | | 100 | $5 \times 10^{-4}$ | $1 \times 10^{-2}$ | 164 | 76.5 |

# B   Image selection using cross entropy

The soft label baseline involves the most naive way of selecting images from training data: random selection. We additionally show that using cross-entropy (CE) as a selection criteria further improves the data quality. For each class, we first use the optimal expert to compute cross-entropy values for each images in the original training set. We then divide the training images into 10 quantiles according to the CE values, where 1 corresponds to "easiest" sample with lowest CE. We perform the CE-selection on TinyImageNet with IPC=1, 10, 50 settings, and report results in Figure 9. Using the "easiest" samples provide a small but consistent improvement ($\approx 1\%$) to random selection, while using the "hardest" samples hurts the performance much more.

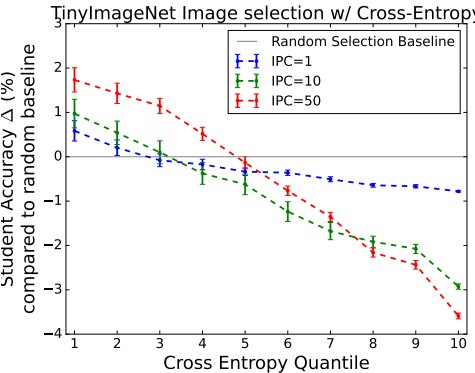

Figure 9: **Selecting images with cross-entropy criteria further improves the soft label baseline performances.** We select training images based on cross-entropy scores using epoch-tuned experts, and report the relative student accuracy change when compared to random selection.

# C   Additional Analysis Results

## C.1   Soft label visualization

We visualize the soft labels for TinyImageNet training set generated by a ConvNet expert at epoch 50 in Figure 10. For every image in the original training set, we use the expert to generate soft labels

and aggregate soft labels by the predicted class (i.e., `argmax`) with simple averaging. The diagonal (probability for the predicted class) is zeroed out so we can see the structures in non-top softmax values. Qualitatively, the clusters present correspond to WordNet ontology [18], some of which we hand annotate with their common ancestors in WordNet.

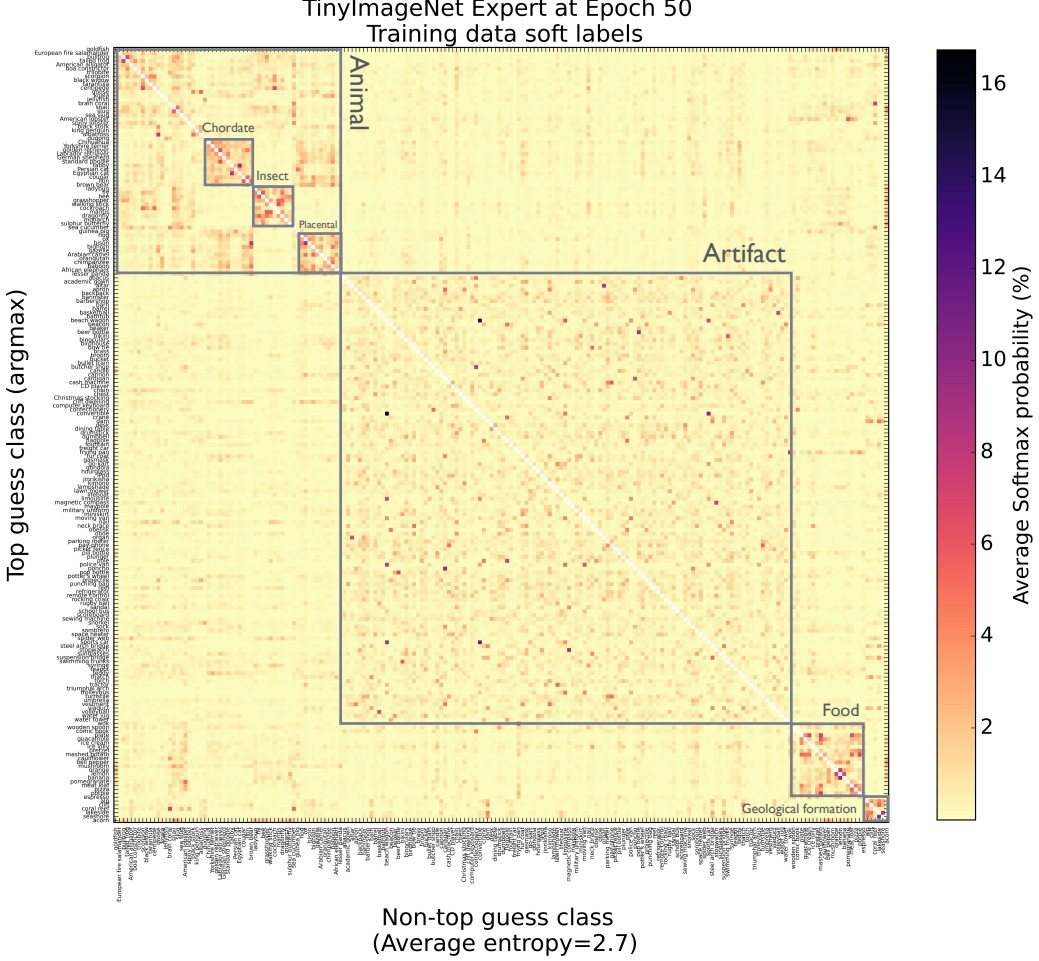

Figure 10: **Visualizing softmax probabilities for each class in Tiny ImageNet** Soft labels are generated by pre-trained experts, and they exhibit structures related to semantic similarity.

## C.2 $i$-th label swapping test

The experiment procedure is the following:

i     Use the optimal early-stopped expert to assign soft labels for a given set of randomly samples training images.

ii     Sort softmax probabilities in descending order

iii     At each iteration $i \in [1, |C|]$, we swap the $i$-th label with the last label.

iv     Train student model on the swapped label, and compute

$$\text{Relative Performance} = \frac{\text{Student Model Accuracy on swapped } i\text{-th label}}{\text{Student Model Accuracy on unswapped label}}$$

## C.3 Data-knowledge Scaling Law

In Figure 11, we repeat the same data-knowledge scaling law in Section 4.3. The comparison Figure 6 and Figure 11 shows that to fully learn the expert model at a later epoch, the student models need more data. For example, to fully learn the expert model at epoch 11 (e.g., Figure 6), with full knowledge, the student model needs less than 20 IPC. In contrast, to fully learn the expert model

at epoch 46 (e.g. Figure 11), the student model needs almost 100 IPC. By combining scaling laws for different experts and for different data budgets, we establish the Pareto frontier for data-efficient learning shown in Figure 6 right.

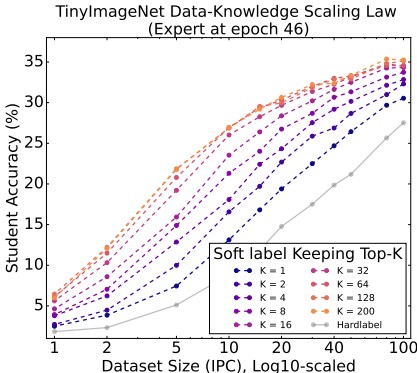

Figure 11: **Empirical Data-Knowledge Scaling Law with expert at epoch 46.** We repeat the scaling law experiment in Section 11 using a later-epoch expert. The comparison between two experiments shows that more data is needed for the student to fully recover a later-epoch expert.

## D    Details on label learning with distillation methods

### D.1    Learn label with MTT

MTT [4] distills data by a trajectory matching objective. Given a network, the method first trains many experts on the full dataset, and saves the entire training trajectory (namely, the expert model parameters at each epoch). To learn the distilled data, a student model is initialized from a random checkpoint of a random expert, and then the student model is trained for several iterations on the distilled dataset. The goal is to match the model parameters (after being trained on the distilled dataset) with the model parameters trained the full dataset at a future epoch. We adopt Algorithm 1 in the original paper [4] but simply freeze images after initializing them with random training data. Labels are initialized as hard labels according to the image class. During distillation, only labels are learned. We perform an extensive hyperparameter search on sensitive parameters such as: $N$: number of steps the student model trains on the distilled data, $M$: the future expert epoch we compare student model parameters to, $T$: the maximum expert epoch we initialize student model parameters with. Best results are reported in Table 3.

Table 3 shows that MTT-adaptation only provides marginal improvements to the hard label baseline. When only images are distilled, the trajectory matching objective is a suitable proxy to the bi-level dataset distillation objective. Nevertheless, the training dynamics of student learning with soft labels could be quite different from the training dynamics of experts learning with hard labels. In other words, when soft labels are used to train student models, it is not guaranteed that matching experts' trajectory remains a suitable proxy for dataset distillation's objective.

### D.2    Learn label with BPTT

We use truncated-BPTT [7, 9] to solve the bi-level optimization problem detailed in Eqn. 1. See Algorithm 1 for details. Instead of learning both images and labels, we initialize images from randomly selected training data and freeze them during distillation and learn label only. The BPTT algorithm has relative high sensitivity to hyperparameters such as $T$: total number of unrolling steps, $M$: truncated window size. Additionally, one common pitfall to BPTT is exploding gradients. To combat the optimization challenge, we adopt the algorithm and the training recipe from Feng et al. [9]. We perform an extensive hyperparameter search and report the best results in Table 3.

The comparison methodology is the following. To compare labels learned by BPTT and those generated from experts, we use the same data ($\{x_i\}_{i=1}^{\text{IPC}}$) to generate labels with either methods. We denote expert-ensemble labels as $\{y_i^l\}$, where $l$ indicates the early-stopped epoch. We denote the BPTT-learned labels as $\{y_i^{\text{BPTT}}\}$. We use Jensen-Shannon Distance (JSD) to quantify the distance between the soft labels generated by two methods.

**Algorithm 1** Learn soft label with BPTT

---

**Require:** Target dataset $\mathcal{D}_{target}$. $T$: total number of unrolling steps. $M$: truncated window size. $\alpha_1$: student network learning rate. $\alpha_2$: distilled data(label) learning rate. $\mathcal{L}$: loss function(cross-entropy for classification)
  Randomly sample images from $\{x_i\}_{i=1}^{\text{IPC}} \sim \mathcal{D}$
  Initialize soft label with hard label. Distilled dataset: $\mathcal{D}_{syn} = \{x_i, y_i\}_{i=1}^{\text{IPC}}$
  **while** Not converged **do**
      Sample a batch of data $d_{target} \in \mathcal{D}_{target}$
      Randomly initialize student network parameter $\theta_0$
      **for** $n : 0 \rightarrow T - 1$ **do**
          **if** If $n == T - M$ **then**
              Start accumulating gradients
          **end if**
          Sample a mini-batch of distilled data $d_{syn} \sim \mathcal{D}_{syn}$
          Perform gradient update on student network parameters $\theta_{n+1} = \theta_n - \alpha_1 \nabla \mathcal{L}(d_{syn}; \theta_n)$
      **end for**
      Compute loss on target data $\mathcal{L}(d_{target}, \theta_N)$
      Perform gradient update on soft label $\{y_i\} \leftarrow \{y_i\} - \alpha_2 \nabla \mathcal{L}(d_{target}, \theta_N)$
  **end while**

---

For each image $x_i$, we compute $\text{JSD}(y_i^{\text{BPTT}}, y_i^l)$ for all $l$. We perform min-max normalization across epochs:

$$\text{Normalized JSD}(y_i^{\text{BPTT}}, y_i^l) = \frac{\text{JSD}(y_i^{\text{BPTT}}, y_i^l) - \min_k \text{JSD}(y_i^{\text{BPTT}}, y_i^k)}{\max_k \text{JSD}(y_i^{\text{BPTT}}, y_i^k) - \min_k \text{JSD}(y_i^{\text{BPTT}}, y_i^k)}$$

In addition to the normalized JSD shown in Figure 8, Figure 12 shows the raw JSD comparisons between BPTT learned labels and ensemble-expert tagged labels. To build a baseline reference, we compute JSD values between soft labels generate by four single experts (only differ by training seeds) on the same set of images. For IPC=1, the JSD between BPTT learned labels and ensemble labels have slightly higher JSDs than the expert pairwise distance. For IPC=10, the JSD values fall into the same distribution. The raw JSD comparison further suggests that BPTT generated labels recovers a softmax distribution that is very similar to ensemble labels.

Similar to standard distillation, BPTT might suffer from the same scaling problem. In other words, it yields suboptimal performance when distilling on larger IPCs. We suspect that optimization might be the cause behind BPTT learned label has worse performance than the best ensemble-expert labels at IPC=10.

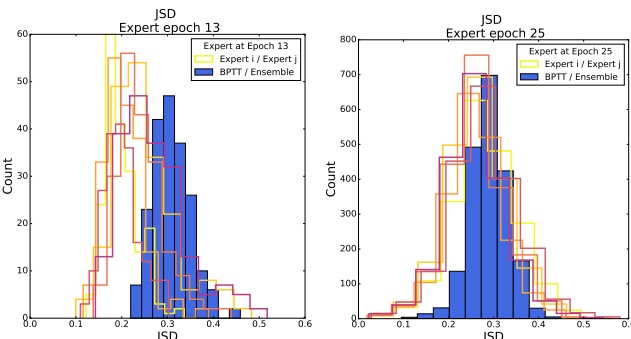

Figure 12: **JSD between BPTT-learned labels and expert-ensemble labels** JSDs between BPTT and expert-ensemble labels fall into the same distribution as JSDs between labels generated by two random experts.

