# OpenReview forum: "A Label is Worth A Thousand Images in Dataset Distillation"
_NeurIPS.cc/2024/Conference — NeurIPS 2024 poster_

### Official Review · Reviewer_Zv9o · 2024-07-08

**Soundness:** 3
**Presentation:** 3
**Contribution:** 4
**Rating:** 7
**Confidence:** 4

**Summary:**

The paper studies the effect of synthetic image soft labels on training performance, and show that the success of DD methods is attributed to the use of informative labels. The authors showed that the structured information in soft labels is important, and there is a tradeoff between knowledge and data. Generally, the paper conducted extensive ablations to analyze the synthetic soft labels.

**Strengths:**

1. The paper studies the interesting and novel aspect of the role of synthetic labels in DD.
2. Thorough experiments are done and the comparisons are fair and meaningful.
3. The experimental results and analysis provide important and interesting insights that are beneficial for the DD community.

**Weaknesses:**

I do not observe major weaknesses of the paper. However, I do have some minor concerns that I would like to discuss with the authors:

1. Observing Figure 4, it is interesting that when swapping the top-1 label with the last one, the relative performance can still be even preserved to relatively 20% and even 50% (IPC=1). The top-1 label should be the correct prediction, and swapping makes all images wrongly labeled. I wonder why there is still 20%~50% relative performance when all the training data are wrongly labeled.
2. For the experiments corresponding to Figure 7 (treatment 2), why no re-normalization? This results in probabilities that do not sum up to 1 and may cause unexpected consequences during training.
3. It seems that there are two types of labels used for analysis. One type is directly downloaded synthetic dataset (image-label pairs) in section 4. Another type is generating soft-labels via ensembling in section 5. Does the second guarantee that the ensembled soft-labels are correct (same one-hot encodings as the original pairs)? Also, using early-stopping experts have drawbacks that it does not provide correct label information (e.g. Figure 3). To what extent (how early) may the experts be useful for label generation?

**Questions:**

See Weaknesses.

**Limitations:**

Yes, limitations are adequately addressed.

---

> ### Author Rebuttal · Authors · 2024-08-05
>
> We are grateful for the reviewer’s constructive comments and are glad they found our work interesting. We respond to their specific comments below:
>
> __Response to Weakness__
> > *Figure 4, regarding performance gain when data are wrongly labeled*
>
> The argmax certainly contains a lot of information. However, our intuition regarding why replacing such an important logit with a wrong one lies in the fact that not all useful information is stored in the top-1 prediction. For example, for IPC=1, the labels are generated with an expert who has only trained for 7 epochs on the real data. As shown in Figure 3, panel 2, the argmax at epoch 7 only has a 2% probability. In fact, the top 5 predictions all share roughly the same likelihood. These top logits contain semantic information such as class feature similarities (see Figure 10 in the Appendix). Therefore, even with the top-1 label being incorrect, the rest of the logits still contain useful and correct information for the model to learn from.
>
> > *For the experiments corresponding to Figure 7 (treatment 2), why no re-normalization?*
>
> It is a perfectly valid concern regarding re-normalization. During our experiment design, we carefully weighed the pros and cons of using normalization for label treatment. The rationale behind not using re-normalization is two-fold.
>
> First, as shown in Figure 2 (right), label entropy significantly impacts final model performance. By using re-normalization, we would alter the label entropy of the treatment group. Second, for each training input, re-normalization only impacts the gradient with a constant scalar shift. Therefore, we believe that whether we perform re-normalization should not matter.
>
> To eliminate the possibility that using un-normalized logits causes downstream effects, we re-ran the treatment 2 group with re-normalization and compared it against the original results (no re-normalization). The results, shown in the attached PDF, confirm that re-normalization does not make a statistically significant difference to the experimental outcomes.
>
>
> > *It seems that there are two types of labels used for analysis. One type is directly downloaded synthetic dataset (image-label pairs) in section 4. Another type is generating soft-labels via ensembling in section 5. Does the second guarantee that the ensembled soft-labels are correct (same one-hot encodings as the original pairs)?*
>
> In Section 3, for the SOTA methods we compare our soft label baselines to, we obtain soft labels directly from the synthetic dataset (i.e., downloaded). In our soft label baseline, the images are randomly sampled from the training data, and labels are generated by experts with epoch tuning.
>
> In Section 5, we further demonstrate that ensembling can bring additional improvements to the soft label baseline. We do not impose the restriction that ensemble soft labels are correct. In other words, no restrictions are imposed to ensure that the argmax of the ensemble labels corresponds to the “correct” class for that particular image.
>
> To our understanding, correctness is less important in the dataset distillation setting, which we discuss further below.
>
>
> > *Also, using early-stopping experts have drawbacks that it does not provide correct label information (e.g. Figure 3). To what extent (how early) may the experts be useful for label generation?*
>
> You have pointed out a very counterintuitive phenomenon! Indeed, early experts generate labels that are “incorrect.” Here, we can loosely define “incorrect labels” as those whose argmax does not lead to a correct classification. In Figure 6 (right), we used experts trained until various stages (from early to late) as the labelers for different data budgets (i.e., different IPC values). We observe that on TinyImageNet, experts as early as Epoch 11 could be useful for label generation under small data budgets (IPC=1). As we increase the data budgets, it becomes more optimal to use later experts.
>
> Our intuition regarding why incorrect labels are more optimal, especially under low data budget regimes, is that when the student model is learning with a limited amount of data, it benefits from mimicking behaviors from a less well-trained teacher, learning only simple features and functions. For example, the student network may learn features/functions that can distinguish between broad categories like dogs (corresponding to many classes in TinyImageNet) and vehicles (also corresponding to many classes in TinyImageNet). Such a crude classifier is the best the model can achieve given the data limit. In other words, with so little data, the student network learns a simpler function because it does not have enough data to learn a complex decision boundary to distinguish between a golden retriever and a German Shepherd. In these “data-limited regimes,” a less well-trained teacher provides better guidance for the student model.

---

> > ### Comment · Reviewer_Zv9o · 2024-08-08
> > **Thank you for the rebuttal**
> >
> > The rebuttal has addressed my concerns. I believe the paper is insightful and meaningful for the DD community. I am keeping my score and vote for acceptance.

---

### Official Review · Reviewer_hrBt · 2024-07-11

**Soundness:** 3
**Presentation:** 4
**Contribution:** 4
**Rating:** 6
**Confidence:** 4

**Summary:**

This paper investigates the importance of soft labels for dataset distillation methods, conducting detailed ablation experiments on the role of labels under various settings. It deeply explores the impact of labels on learning, providing an in-depth analysis and study of the intrinsic properties of labels. The work also provides empirical scaling laws that characterize the effectiveness of soft labels as a function of images-per-class in the distilled dataset and establishes an empirical Pareto frontier for data-efficient learning.

**Strengths:**

1. This paper clearly demonstrates the crucial role that soft labels play in the effectiveness of data distillation methods. This aspect has never been carefully studied in previous dataset distillation work; it was usually considered an additional trick and did not receive much attention. This work points out new directions for data-efficient learning.
2. The designed experiments are interesting and comprehensive, presenting what constitutes good soft labels, the importance of knowledge (soft labels) in learning, and how to effectively obtain higher-quality soft labels.

**Weaknesses:**

1. According to Table 7, the expert model (epoch) used to produce soft labels in the soft label baseline appears to be carefully selected. Does the choice of epoch introduce additional costs? Additionally, do the other mentioned dataset distillation methods also use the best soft labels from these epochs to ensure a fair comparison?
2. MTT and SRe2L are not the SOTA methods currently. I would like to know if more advanced methods like DATM [1] and G-VBSM [2] still heavily rely on soft labels and whether their performance still can not surpass the soft label baseline.
Minor: line 185 In Figure 3 left.

[1] Towards lossless dataset distillation via difficulty-aligned trajectory matching. ICLR 2024

[2] Generalized large-scale data condensation via various backbone and statistical matching. CVPR 2024

**Questions:**

Please refer to weaknesses.

---

> ### Author Rebuttal · Authors · 2024-08-05
>
> We are grateful for the reviewer’s constructive comments and thoughtful insights. We respond to their specific comments below:
>
> __Response to Weakness__
>
> > *According to Table 7, the expert model (epoch) used to produce soft labels in the soft label baseline appears to be carefully selected. Does the choice of epoch introduce additional costs?*
>
> Yes, tuning the expert epoch is used for our soft label baseline, and it does come at a cost. Other SOTA methods spend the majority of compute on generating images. For example, Ra-BPTT requires more than 100 GPU hours to generate distilled images for the results shown in Figure 1. In contrast, the soft label baseline simply uses randomly selected training images, incurring no costs on image generation or selection.
>
> For methods like FrePo and SRe2L, generating the best images also requires extensive hyper-parameter tuning in their distillation pipelines. Therefore, we do not believe the additional cost incurred for epoch tuning will be the bottleneck.
>
> Table 7 might give the impression that the epoch needs to be “carefully” selected. However, we hope these hyperparameters will ensure easy reproducibility. In Figure 6 (right), we showcase how to choose the “optimal expert epoch” by establishing a Pareto frontier. From this figure, our understanding is that we can establish a robust Pareto front with only five expert epochs, covering the optimal expert for IPC values ranging from 1 to 200.
>
>
> > *Additionally, do the other mentioned dataset distillation methods also use the best soft labels from these epochs to ensure a fair comparison?*
>
> Thank you for the question! We address this concern in the global response with new plots included in the PDF attachment.
>
> > *MTT and SRe2L are not the SOTA methods currently. I would like to know if more advanced methods like DATM [1] and G-VBSM [2] still heavily rely on soft labels and whether their performance still can not surpass the soft label baseline.*
>
> The field has been advancing rapidly. DATM introduces innovation to MTT based on “difficulty-level” trajectory matching. They claim that by using late trajectories, they improve performance for larger synthetic sets (i.e., large IPC) and achieve lossless distillation (i.e., matching distilled data to the quality of training data). Section 4.3 in DATM shows that they also heavily rely on soft labels. For TinyImageNet, they achieve 39.7% “lossless” test performance with IPC=50. Our soft label baseline achieves 35.6%. While the soft label baseline underperforms compared to DATM, the comparison still reflects the great importance of labels.
>
> Similarly, reading Section 4.1in G-VBSM, our understanding is that G-VBSM also heavily relies on soft labels. Several improvements they leverage, including ensemble techniques, align with our observations. Since G-VBSM builds on SRe2L and the soft label baseline performs on par with SRe2L, G-VBSM is able to achieve further improvements.
>
> We do not aim to outperform all SOTA methods using such a simple baseline, instead we hope to bring some understanding of what information is being distilled that leads to data-efficient learning and correct the misconception that labels are less important compared to images in dataset distillation.

---

> > ### Comment · Reviewer_hrBt · 2024-08-09
> > **Thanks for the reply**
> >
> > I have read this paper and the author rebuttal (regarding all the reviewers' questions) thoroughly. I appreciate the contribution of this work and keep my score.

---

### Official Review · Reviewer_UKPz · 2024-07-13

**Soundness:** 3
**Presentation:** 2
**Contribution:** 3
**Rating:** 5
**Confidence:** 4

**Summary:**

This paper introduces soft probabilistic labels to the dataset distillation task. Specifically, it finds that the labels should consider structured information and perform unequally. Experiments on diverse datasets demonstrate its effectiveness.

**Strengths:**

1) This paper proposes the introduction of soft labeling in dataset distillation and provides an interesting analysis of the intrinsic properties of data distillation, rather than focusing on improving the base module of specific methods.
2) The method is well-supported, and the results are reliable.

**Weaknesses:**

1) The motivation needs to be stated more clearly. Why are label-level methods regarded as superior to image-level methods？
2) In the introduction, the author introduces several techniques such as 'expert' models, Pareto frontier, and knowledge-data scaling laws but lacks detailed explanation.
3) Section 3.2 requires reorganization to enhance readability.
4) The structure information needs to be expressed more clearly and intuitively.

**Questions:**

1) In the Method section, could different soft-labeling strategies (excluding cutmix) influence the entire pipeline?
2) Does early stopping result in varied performances across different baselines and datasets? How do the authors address this issue?
3) In the experiment, the author uses the swap test to validate the structure information and claims “Top labels contain structured information and the non-top contain unstructured noise”. Why not just remove the unstructured noise?

**Limitations:**

1) The writing requires improvement as the author attempts to cover too information without adequately establishing connections and justifying the necessity of the key technologies.
2) The tables and figures need reorganization to enhance clarity. Reviewer currently struggles to understand their intended messages.
3) Some typos need to be corrected, such as those in the representation of Figure 4.
4) The limitations mentioned in the Conclusion are not clearly stated.

---

> ### Author Rebuttal · Authors · 2024-08-05
>
> Thank you for the constructive feedback! We respond to their specific comments below:
>
> __Response to Weakness__
>  > *The motivation needs to be stated more clearly. Why are label-level methods regarded as superior to image-level methods?*
>
> To our best understanding, the dataset distillation community does not consider label-level methods superior to image-level methods. In fact, it appears that more work has been dedicated to image-level methods, often treating labels as a minor additional “trick” to boost performance.
>
> The main motivation behind our study is to emphasize the importance of soft probabilistic labels in the dataset distillation task. We aim to correct this misconception and argue that labels are, in fact, quite central to data distillation. Overall, we do not believe that label-level methods are necessarily superior to image-level methods. We hope our work inspires future research to explore methods that effectively utilize both images and labels to achieve the best outcomes.
>
> > *In the introduction, the author introduces several techniques such as 'expert' models, Pareto frontier, and knowledge-data scaling laws but lacks detailed explanation.*
>
> Thank you for the constructive feedback! Our introduction will indeed be clearer with a more detailed explanation of key terms when we first introduce them. We will incorporate your suggestions in the camera-ready version for better readability. We also included a detailed explanation in Official comment.
>
> > *The structure information needs to be expressed more clearly and intuitively.*
>
> We define structured information as the softmax values in logits that contain semantic information, such as class or feature similarities. These are also referred to as “dark knowledge” [1] or “supervisory signals” [2] by the knowledge distillation community. For example, a soft label for a goldfish image may assign a 70% likelihood to the “goldfish” class and a 10% likelihood to the “orange” class. The 10% likelihood assigned to “orange” is not noise; it indicates that the two classes are similar, likely due to color. We consider these logits to contain “structured noise.” We expect this pattern (goldfish-orange) to appear consistently in many goldfish images.
>
> Conversely, if the model randomly assigns a 2% likelihood to the “pizza” class due to spurious features or other noise, we do not expect all goldfish images to have a consistent pattern of being slightly identified as pizza. We consider this as “unstructured noise.”
>
>
> __Response to Questions__
>
> > *In the Method section, could different soft-labeling strategies (excluding cutmix) influence the entire pipeline?*
>
> Yes, different labeling strategies can impact the distilled outcome. In this paper, we employ a simple labeling strategy, which involves using experts trained at different epochs as the labelers to generate labels. Specifically, our soft label baseline uses this expert-based soft labeling strategy on randomly selected training images. We demonstrate that this simple baseline achieves performance comparable to state-of-the-art distillation results.
>
> In Section 5, we explore a few alternative labeling strategies and show how these strategies impact the distillation pipeline and outcomes, still within the setting where images are randomly selected from the training data. We hope that future work will further explore and develop labeling strategies to improve the dataset distillation pipeline.
>
> > *Does early stopping result in varied performances across different baselines and datasets? How do the authors address this issue?*
>
> Based on the dataset and the data budget (IPC), the optimal early stopping epoch varies, as shown in Figure 6 (right). In this figure, we vary both the expert epoch and the data budget. For larger data budgets (IPCs), it is more effective to use a later epoch to generate labels. From this observation, we understand that the optimal information stored in soft labels can differ significantly depending on the data budget.
>
> We believe it would be an interesting follow-up for future work to study how one can predict the optimal epoch given a data budget for different datasets. In this work, we aim to shed light on the fact that early stopping plays an important role in determining the quality of labels. For instance, in Figure 2 (left), we demonstrate this effect on ImageNet-1K.
>
> > *In the experiment, the author uses the swap test to validate the structure information and claims “Top labels contain structured information and the non-top contain unstructured noise”. Why not just remove the unstructured noise?*
>
> Indeed, we completely agree that this is a valid suggestion. By removing unstructured noise, we might further improve the performance of our soft label baseline. In our analysis, we also identified additional modifications to our soft label baseline for boosting performance, such as ensembling (Section 5.1) and temperature smoothing (Section 4.2). Practitioners could benefit from these additional enhancements, potentially including the removal of unstructured noise.
>
> In our reported soft label baseline, we did not include any of these add-ons because we aimed to keep our baseline as simple as possible. Our goal was to demonstrate that soft labels are crucial for dataset distillation.
>
>
> __Response to Limitations__
>
> > The limitations mentioned in the Conclusion are not clearly stated.
>
> Thank you for pointing out that our work could benefit from a more in-depth exploration of limitations. We appreciate this valuable feedback. Here is a brief version of what we intend to add to our conclusion to further address these limitations:
>
> > Writing and Presentation
>
> We will carefully revise writing and presentation for the camera ready version.
>
> [1] Distilling the Knowledge in a Neural Network https://arxiv.org/abs/1503.02531
>
> [2] Rethinking soft labels for knowledge distillation: A Bias-Variance Tradeoff Perspective https://arxiv.org/pdf/2102.00650

---

> ### Author Response · Authors · 2024-08-05
> **Glossary of Key Terms**
>
> __Expert Models__: Also referred to as teacher models, these are models that have been trained on the original training data and are used to generate soft labels.
>
> __Pareto Frontier__: In the context of dataset distillation, the objective is to optimize for model performance and data budgets. The Pareto frontier represents the set of points where each point corresponds to the best model accuracy achievable for a given data budget. One cannot achieve better model performance with the same data budget.
>
> __Knowledge-Data Scaling Laws__: Data scaling laws describe how the performance of a model (e.g., measured by test accuracy) improves as a function of dataset size. We propose knowledge-data scaling laws to describe how the use of expert knowledge (i.e., soft labels) can shift the standard scaling law.

---

> ### Author Response · Authors · 2024-08-05
> **Limitations**
>
> __Soft Label Baseline__: We have highlighted the importance of soft labels using a simple soft label baseline. We leave it for future work to explore the best ways to optimize both labels and images during distillation, and to study how each can impact student learning in different ways. Additionally, future work can investigate what other information, beyond expert knowledge, can be distilled to achieve data compression.
>
> __Label Generation Strategy__: We have explored label generation strategies based on existing methodologies, including using pretrained experts and Ra-BPTT. We believe future research could further explore optimal ways to generate more informative labels.
>
> __Data Modality__: Similar to most dataset distillation work, we have primarily focused on image classification tasks. While we believe our conclusions can generalize to other data modalities, a limitation of this work is the diversity of tasks explored.

---

### Official Review · Reviewer_pAYe · 2024-07-18

**Soundness:** 3
**Presentation:** 3
**Contribution:** 3
**Rating:** 6
**Confidence:** 4

**Summary:**

This paper analyses the role of soft labels used in dataset distillation. Experiments with different ablation studies show that the performance of soft labels based data distillation approaches is primarily attributed to the use of soft labels. Secondly, the authors study the various types of soft labels and their effect on model performance. Additionally an empirical scaling law is provided to characterize the relation between effectiveness of soft labels and image per class in distilled dataset.

**Strengths:**

The paper is written clearly with well-presented motivation and is easy to follow

Extensive analysis and ablations are presented providing a better understanding of role of labels in dataset distillation

The paper focuses on a largely overlooked aspect of dataset distillation methods: the degree of contribution of soft labels.

**Weaknesses:**

The paper could benefit from a theoretical analysis of why soft labels are so effective. Additionally, the generalizability of the findings to data distillation in other modalities could be insightful.

Minor comments:

1. The description in line 222 appears inconsistent with results in figure 4. IPC=1 appears more robust to swapping of top labels compared to IPC=10

2. A few writing and grammar errors exist in related work, line numbers 87-90. Also, 'generation' should be 'general' in line  167

3. In Figure 2, it might be better to use the same dataset to highlight the dependence on expert accuracy and label entropy.

**Questions:**

I am concerned about the use of experts at tuned epoch for comparison of soft label baseline with previous methods. Are the expert epochs also tuned for previous approaches?

Can the authors provide insight as to why structured information is more beneficial in soft labels in the case of dataset distillation while the opposite is true for knowledge distillation as mentioned in related work?

 It is unclear how past approaches in figure 1 right  (which as I understood to originally use soft labels) were adapted to hard labels. Is it done by performing argmax on the soft labels?

---

> ### Author Rebuttal · Authors · 2024-08-05
>
> Thank you for the constructive feedback! We are glad that you found our work interesting. We respond to their specific comments below:
>
> __Response to weakness__
>
> > *The paper could benefit from a theoretical analysis of why soft labels are effective. The generalizability of the findings to data distillation in other modalities could be insightful.*
>
> Soft labels are effective because compared to their hard label counterparts, they contain probability for each class for a given image. Those logits not only convey the class information for the given image but also contain other information such as class similarities. While it can be challenging to study the theoretical framework for a large ConvNet on an image classification task, we believe one can demonstrate the effectiveness for smaller convex optimization problems, such as simple regression. Specifically, one can show how providing logits can reduce sample complexity to learn the same classifier. We believe formalizing this form from a learning theory persepective is an exciting avenue for future work.
>
> To date, almost all dataset distillation work has focused exclusively on image classification tasks. To our knowledge, the only work extending this line of research beyond image classification is [1], where the authors explored dataset distillation for vision-language models. We believe that the conclusions drawn from our work, particularly the importance of labels, should apply to other vision tasks and classification tasks in other modalities. As future dataset distillation research begins to explore other modalities, we hope the soft label baseline we established will serve as a strong starting point for future work. We acknowledge that our work could benefit from additional results in other modalities, and we are currently considering including language modeling tasks for the camera-ready version.
>
> > *The description in line 222 appears inconsistent with results in figure 4. IPC=1 appears more robust to swapping of top labels compared to IPC=10*
>
> Thank you for pointing out the mistake. Indeed, the description in line 222 is not entirely accurate. If we only consider $i=1$ in the "Swap i-th label" test, IPC=1 is more robust to swapping top labels compared to IPC=10. However, when we consider $2≤i≤32$ (swapping top-k labels for  $k>1$), IPC=1 relies more on top-k labels than IPC=10, and thus is less robust to swapping top-k (k>1) labels than IPC=10. We will ensure this clarification is made in the camera-ready version.
>
> __Response to questions__
> > *I am concerned about the use of experts at tuned epoch for comparison of soft label baseline with previous methods. Are the expert epochs also tuned for previous approaches?*
>
> Thank you for the question! We address this concern in the global response with new plots included in the PDF attachment.
>
> > *Can the authors provide insight as to why structured information is more beneficial in soft labels in the case of dataset distillation while the opposite is true for knowledge distillation as mentioned in related work?*
>
> You have pointed out a very interesting contradiction! Our understanding is that the key difference between data distillation and knowledge distillation lies in data budgets. In knowledge distillation, the student network has full access to the entire training dataset, along with soft labels generated by the expert (teacher). Conversely, in the data distillation setting, we impose a strict limitation on the size of the data. As a result, the student network must rely more heavily on the labels generated by the teacher. Therefore, it relies more on structured information in data-poor settings.
>
> > It is unclear how past approaches in figure 1 right (which as I understood to originally use soft labels) were adapted to hard labels. Is it done by performing argmax on the soft labels?
>
> For SRe2L, MTT, and FRePo, the distilled images are initialized with real images from the training data. To obtain hard labels, we use the label based on the “intialization” image. Specifically, in MTT (TESLA) and FRePo, the original works include comparisons of hard versus soft labels, and they both chose to use hard labels in the same way (i.e., hard labels based on the initialization image). We maintained the choice made by the original authors for consistency and reproducibility. However, we suspect that the outcome would be the same if we used the argmax of the soft labels to obtain hard labels, as each distilled image is supposed to be representative of its respective class.
>
> For Ra-BPTT, where images are initialized with random Gaussian noise, we use the argmax of the soft labels as the hard labels.
>
> [1] Vision-Language Dataset Distillation https://arxiv.org/abs/2308.07545

---

### Author Rebuttal · Authors · 2024-08-06

We want to thank all the reviewers for the detailed and thoughtful response! We are glad that reviewers have found our work interersting and scientifically sound. We have carefully read through all the comments and we believe that all your feedback will bring improvements to our work!

We address some common questions below.

__Soft label generation with epoch tuning__

Among all the reviews, we noticed a common theme regarding experiment details for Figure 1, where we compare our soft label baseline with SOTA methods. Specifically, there are questions regarding how soft labels are generated for SOTA methods. We provide further details below, and the new experiment results can be found in the attached PDF.

We have compared our soft label baselines to four existing methods, each with their own soft label generation strategies proposed by the original authors. In Figure 1, we applied the original soft label generation strategy used by each method. Overall, some of these methods already include epoch tuning (MTT/TESLA), while others do not (SRe2L), and for some, the concept of epoch tuning is not well-defined (Ra-BPTT and FRePo).

To address your concern, we have also included a version of Figure 1 where we apply the exact same label generation strategy used for the soft label baseline (expert label generation with epoch tuning) on SRe2L. Further analysis is provided in the supplementary PDF.

We clarify how soft labels are obtained for each of the methods below:

* SRe2L: The original method uses labels generated from a fully trained expert without epoch tuning. Therefore, in our reported results for SRe2L, the soft labels are not epoch-tuned.

* MTT (TESLA): In the original method, the expert used to generate labels is already epoch-tuned, and the epoch also impacts the image generation pipeline.


* Ra-BPTT: The original method generates labels along with images using a bi-level optimization objective (Eq 1), so no experts are trained during the process. As labels are not generated by experts, epoch tuning is not applicable. We experimented with using pre-trained experts to generate labels for Ra-BPTT generated images but observed that experts trained on real images perform poorly on Ra-BPTT generated data. This is likely because the generated images are too out-of-distribution for experts trained on real training data. Thus, the original labels generated by Ra-BPTT should be considered optimal for this method.


* FRePo: Similar to Ra-BPTT, FRePo is based on Back-Propagation in Time (BPTT), and no experts are trained during the distillation process. Like Ra-BPTT, labels are learned during the distillation process along with images.

We included a revised version of Figure 1 in the PDF. Our additional results show that (1) soft label strategy used by the original method is more effective for the given method than epoch tuning (2) our soft label baseline remains competitive. Please refer to the supplementary PDF for further details.

__Figure 7 revision__
In addition, we included a revised version of Figure 7 (data-efficient experiment with zero shot learning). In this new figure, we address the concerns regarding the softmax re-normalization raised by reviewer Zv9o.Please refer to the supplementary PDF for further details.

---

### Decision · Program_Chairs · 2024-09-25

**Decision:**

Accept (poster)

**Comment:**

The manuscript has been reviewed by four reviewers. All of the reviewers, after rebuttal, gave positive ratings (including borderline positive).

Essentially, the reviewers found the study well-motivated, clearly written, and sufficiently verified by experiments.

The AC agrees with the consensus, and recommends the manuscript to be accepted.

Congrats!